# Intracerebral mechanisms explaining the impact of incidental feedback on mood state and risky choice

Romane Cecchi[1], Fabien Vinckier[2], Jiri Hammer[3], Petr Marusic[3], Anca Nica[4], Sylvain Rheims[5], Agnès Trebuchon[6,7], Emmanuel J Barbeau[8,9], Marie Denuelle[9,10], Louis Maillard[11], Lorella Minotti[12], Philippe Kahane[12], Mathias Pessiglione[11], Julien Bastin[1]*

[1]Grenoble Institut Neurosciences, University Grenoble Alpes, Grenoble, France; [2]Motivation, Brain and Behavior (MBB) Team, Paris Brain Institute, Pitié-Salpêtrière Hospital, Paris, France; [3]Université de Paris, Paris, France; [4]Department of Psychiatry, Service Hospitalo-Universitaire, GHU Paris Psychiatrie & Neurosciences, Paris, France; [5]Neurology Department, 2nd Faculty of Medicine, Charles University, Motol University Hospital, Prague, Czech Republic; [6]Neurology Department, University Hospital of Rennes, Rennes, France; [7]Epileptology Department, Timone Hospital, Public Assistance Hospitals of Marseille, Marseille, France; [8]Functional Neurology and Epileptology Department, Hospices Civils de Lyon and Université de Lyon, Lyon, France; [9]Centre de recherche Cerveau et Cognition, Toulouse, France; [10]Neurology Department, CHU Toulouse, Toulouse, France; [11]Neurology Department, University Hospital of Nancy, Nancy, France; [12]Neurology Department, University Hospital of Grenoble, Grenoble, France

*For correspondence:
julien.bastin@univ-grenoble-alpes.fr

Competing interest: The authors declare that no competing interests exist.

**Abstract** Identifying factors whose fluctuations are associated with choice inconsistency is a major issue for rational decision theory. Here, we investigated the neuro-computational mechanisms through which mood fluctuations may bias human choice behavior. Intracerebral EEG data were collected in a large group of subjects (n=30) while they were performing interleaved quiz and choice tasks that were designed to examine how a series of unrelated feedbacks affect decisions between safe and risky options. Neural baseline activity preceding choice onset was confronted first to mood level, estimated by a computational model integrating the feedbacks received in the quiz task, and then to the weighting of option attributes, in a computational model predicting risk attitude in the choice task. Results showed that (1) elevated broadband gamma activity (BGA) in the ventromedial prefrontal cortex (vmPFC) and dorsal anterior insula (daIns) was respectively signaling periods of high and low mood, (2) increased vmPFC and daIns BGA respectively promoted and tempered risk taking by overweighting gain vs. loss prospects. Thus, incidental feedbacks induce brain states that correspond to different moods and bias the evaluation of risky options. More generally, these findings might explain why people experiencing positive (or negative) outcome in some part of their life tend to expect success (or failure) in any other.

## Editor's evaluation

This study used intracranial EEG to explore links between broad-band γ oscillations and mood, and how they impact risky decision-making. The topic is interesting for cognitive neuroscientists and researchers interested in computational psychiatry.

## Introduction

Humans often make inconsistent decisions, even when facing seemingly identical choices. This surprising variability is a major difficulty for rational decision theory and a reason for introducing stochastic functions in choice models. However, even if stochastic functions help mimicking behavior on average, they cannot help predicting individual choice. For this, one needs to identify factors whose fluctuations are systematically associated with changes in preference, in order to replace randomness with bias. Influent factors can be related to the internal state of the decision maker and/or the external context of the choice situation. In neuroscience, many studies have shown that brain activity preceding the presentation of choice options might provide a bias on the eventual choice, particularly when alternatives are safe vs. risky options (*Chew et al., 2019*; *Huang et al., 2014*; *Kuhnen and Knutson, 2005*; *Lopez-Persem et al., 2016*; *Padoa-Schioppa, 2013*; *Vinckier et al., 2018*).

In some cases, baseline brain activity could be related to mental constructs such as pleasantness, satiety, or mood, which were themselves under the influence of external events (*Abitbol et al., 2015*; *Vinckier et al., 2018*). These findings therefore provide a putative mechanism explaining why mood is predictive of risky choice: positive/negative events that increase/decrease mood at the mental level also modulate baseline activity at the neural level, changing the way dedicated brain regions evaluate risky options. More precisely, good mood would lead to overweighting gain prospects and bad mood to overweighting loss prospects, making the overall expected value of a risky option positive or negative, depending on circumstances. Such a mechanism could account for the well-documented impact of mood on buying lottery tickets or investing in financial markets (*Bassi et al., 2013*; *Otto et al., 2016*; *Saunders, 1993*), an effect that has been reproduced in the lab (*Arkes et al., 1988*; *Chou et al., 2007*; *Eldar and Niv, 2015*). It could also account for why depressed and manic patients have opposite attitudes toward risk, respectively, focusing on negative and positive outcomes of potential actions (*Leahy, 1999*; *Yuen and Lee, 2003*).

Despite the importance of this phenomenon, producing disastrous decisions at both the individual scale in psychiatric conditions and the societal scale in real-life economics, the underlying mechanism at the neural level is still poorly understood. This is due to the limitations of functional MRI (fMRI), which has been mostly used because mood fluctuations are difficult to track in animals, while invasive techniques are forbidden in humans for obvious ethical reasons. There is nonetheless a particular clinical situation that offers the opportunity to record intracerebral EEG (iEEG) activity from deep electrodes, when patients with refractory epilepsy are implanted prior to surgery. These iEEG recordings have already been used to successfully decode mood from resting-state activity, a technical achievement that may open the route to closed-loop stimulation procedures (*Bijanzadeh et al., 2013*; *Sani et al., 2018*). They have also been used to specify the impact of lateral orbitofrontal cortex stimulation aiming at improving mood in moderately depressed patients (*Rao et al., 2018*). There is however a discrepancy between these pioneering iEEG studies, which reported that changes in mood states might correspond to changes in the frequency of oscillatory activity (*Bijanzadeh et al., 2013*; *Kirkby et al., 2018*; *Rao et al., 2018*), and fMRI studies that have related mood fluctuations to the relative activity of opponent brain systems associated to either reward or punishment processing (*Vinckier et al., 2018*).

To examine whether good and bad mood are associated to activity in different brain regions or different frequency bands, we recorded iEEG activity while patients were performing a task similar to that used in a previous fMRI study to induce mood fluctuations and test their impact on risky choices (*Vinckier et al., 2018*). Mood fluctuations were induced by feedbacks provided to subjects on their responses to general knowledge questions. Choices were about whether to accept a challenge rewarded with monetary gains in case of success but punished with monetary losses in case of failure. To avoid repeating self-report too frequently, a computational model was developed, building on previous suggestion (*Eldar and Niv, 2015*), to generate mood level as an integration of past feedbacks, the perception of which was itself modulated by mood level. This theoretical mood level (TML) was positively correlated with activity in brain regions classically associated with reward, such as the ventromedial prefrontal cortex (vmPFC), and negatively correlated with regions associated with punishment, such as the anterior insula (aIns). In turn, baseline activity in these regions (prior to the presentation of choice options) was predicting a bias on choice, relative to an expected utility model. Indeed, high vmPFC activity favored the risky option (accepting the challenge) by increasing the weight on potential gain, whereas high aIns activity favored the safe option (declining the challenge)

by increasing the weight on potential loss. Yet, due to the poor temporal resolution of fMRI, there was a remaining gap in the explanation regarding the contribution of activity in mood-related regions during decision making. Here, leveraging the excellent temporal resolution of iEEG, we intended to separate brain activity related to feedback events, to the overall mood level, and to the choice process.

Thus, in an attempt to clarify the neuro-computational mechanism through which mood fluctuations might arise and bias decisions under risk, we collected iEEG data in a large group of subjects (n=30) while they were performing interleaved quiz and choice tasks. The objectives of our analytical approach were (1) to identify brain regions in which activity was related to both mood state (good or bad) and choice (safe or risky), and (2) to examine whether mood fluctuations and related decisions were associated to a shift in frequency bands or in anatomical location of oscillatory iEEG activity.

## Results

The aim of this study was to elucidate how intracerebral activity may underpin the impact of mood fluctuations on risky choices. iEEG data were collected in 30 patients suffering from refractory epilepsy (39.5±1.9 years old, 14 females, see demographic details in Table S1 in *Supplementary file 1*), while they performed two unrelated but interleaved tasks. The first was a quiz task designed as a mood induction procedure, and the second was a choice task used to unravel the effects of mood fluctuations on economic choices (*Figure 1a*). In the quiz task, subjects were asked to answer general knowledge questions and received feedback on their response (correct, incorrect, or too late). In order to predictably modulate mood, the difficulty of questions and the validity of feedbacks were manipulated, unbeknownst to the subjects, so as to create episodes of high and low correct feedback rate (see Materials and methods, and *Figure 1—figure supplement 1*). In 25% of the trials, the effect of feedback history was assessed by asking subjects to rate their mood on a visual analog scale. Otherwise (in 75% of trials), the quiz and choice task were separated by a 4-s rest period (black screen). In the choice task, subjects had to decide whether to accept or reject a given offer before performing a challenge consisting in stopping a moving ball inside a target. The offer comprised a gain prospect (i.e. the amount of money they would win in case of success), a loss prospect (i.e. the amount of loss in case of failure), and the difficulty of the upcoming challenge (target size). When accepting the offer, subjects played for the proposed amounts of money, otherwise, when declining, they played for minimal stakes (winning or losing 10 cents). All three choice dimensions (gain, loss, and difficulty) were varied on a trial-by-trial basis. In order to avoid learning and additional effects on mood, no feedback was provided regarding performance in the challenge (actual success rate was around 30% on average but significantly varied with difficulty). Subjects were explicitly informed that the tasks were independent, such that responses to the quiz or mood ratings had no influence on choice options and hence on their monetary earnings.

### Choice behavior

We first tested whether subjects performed the choice task correctly by checking that the three dimensions of the offer (gain prospect, loss prospect, and challenge difficulty) were properly integrated into their choices (*Figure 1b*). Choice acceptance was fitted at the individual level using a logistic regression model that included the three dimensions, and significance of regression estimates was assessed at the group level (across n=30 subjects) using two-sided, one-sample, Student's t-tests. Results show that acceptance rate significantly increased with gain ($\beta_{gain}$ = 0.12 ± 0.02, t[29] = 6.85, and p=$2.10^{-7}$) but significantly decreased with loss ($\beta_{loss}$ = –0.10 ± 0.02, t[29] = –5.51, and p=$6.10^{-6}$) and difficulty ($\beta_{diff}$ = –0.03 ± 0.01, t[29] = –2.97, and p=$6.10^{-3}$). These results were expected given previous behavioral evidence obtained with a similar choice task that subjects can integrate these three dimensions when making a decision (*Vinckier et al., 2018*).

We next assessed whether mood fluctuations impacted choices above and beyond the influence of these three dimensions. We thus tested the link between mood ratings and choice residuals once the three dimensions had been regressed out (*Figure 1c*). To account for interactions and non-linearities in a principled way, we fitted a model based on expected utility theory, which was previously shown to best capture choices in this task (*Vinckier et al., 2018*). In brief (see Materials and methods for details), acceptance likelihood was estimated as a softmax function of expected utility, calculated as

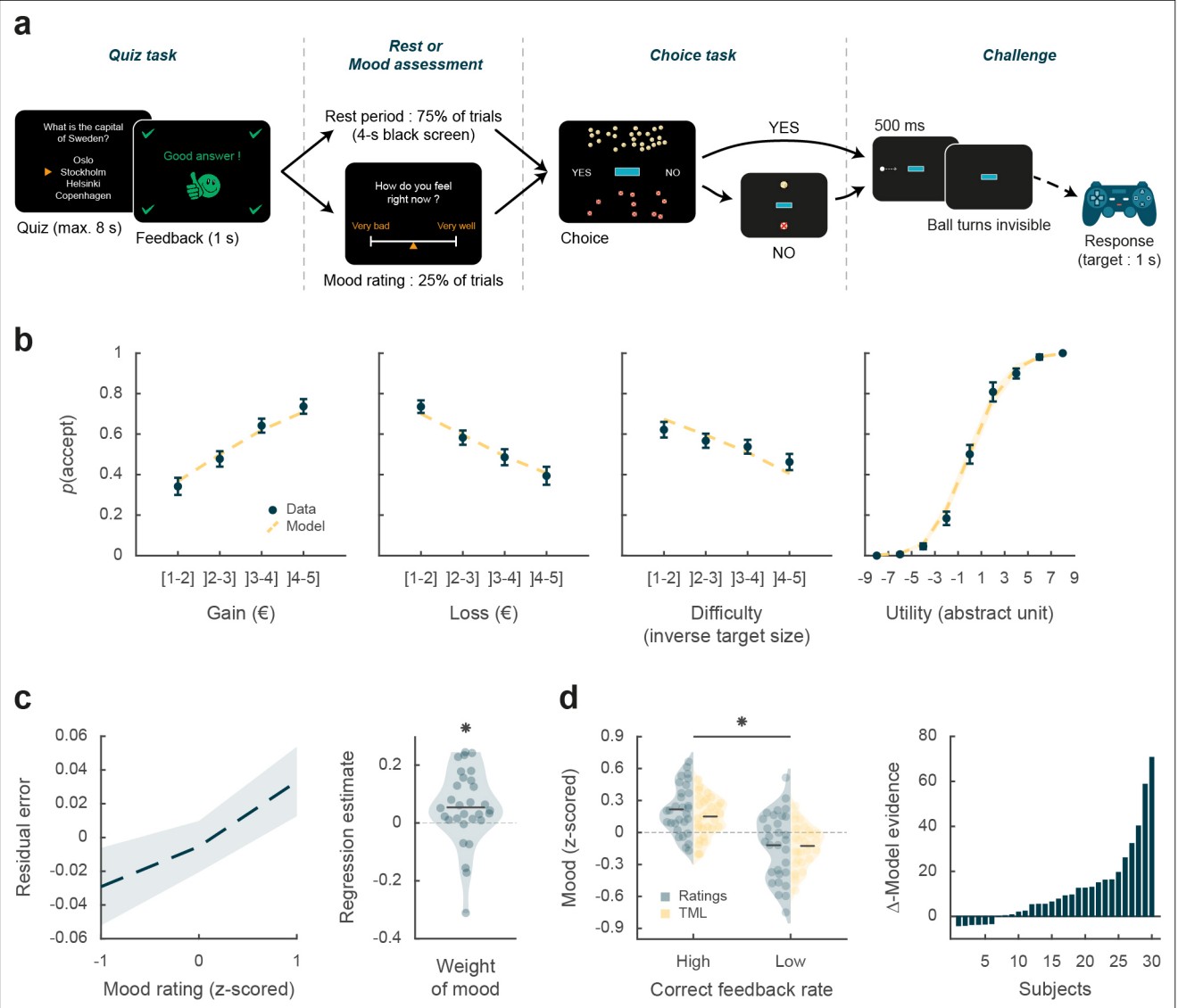

**Figure 1.** Behavioral task and results. (**a**) Trial structure. Each trial included a quiz task, a rest or mood assessment period, and a choice task followed by a challenge. In the quiz task, subjects answered a general knowledge question and received a feedback. The quiz task was followed by a rest period (75% of trials) or a mood rating task (25% of trials) on a visual analog scale. In the choice task, subjects had to decide whether to accept or reject a given challenge by taking into account gain prospects (represented by a bunch of regular 10-cent coins), loss prospects (crossed out 10-cent coins), and difficulty (inversely proportional to the size of the blue bar in the middle of the screen). The challenge consisted in stopping a moving ball, invisible when inside the blue target. (**b**) Choice behavior. Acceptance probability is plotted as a function of the three objective dimensions (gain, loss, and difficulty) and modeled subjective utility of the proposed challenge. Circles are binned data averaged across subjects. Yellow dotted lines represent acceptance probability as computed by the choice model. Error bars represent inter-subject SEM (n = 30). (**c**) Impact of mood on the choice model residual error (actual choice – modeled acceptance probability). Left panel: residual error is plotted as a function of mood rating. Right panel: the weight of mood on residual error is shown as individual regression estimates. Circles represent individual data, and horizontal line represents mean across subjects (as in d, left panel). (**d**) Mood fluctuations. Left panel: effect of correct feedback rate on mood rating and theoretical mood level (TML). Right panel: difference in model evidence between TML and a null model in which feedback had no impact on mood. Bars show subjects ranked in ascending order. Stars indicate significance (p<0.05) using two-sided, one-sample (c) or paired (d), Student's t-test.

The online version of this article includes the following figure supplement(s) for figure 1:

**Figure supplement 1.** Variations in mood rating (black dots) and theoretical mood level (TML, yellow line) across all trials of an experimental session for a single subject, superimposed with episodes of high (blue) and low (red) positive feedback rates.

potential gain multiplied by probability of success ($p_s$, inferred from target size) minus potential loss multiplied by probability of failure ($1 - p_s$), with gain and loss terms weighted by distinct parameters ($k_g$ and $k_l$, respectively). Residuals of this choice model were then regressed against mood ratings at the individual level, and significance of regression estimates was assessed at the group level, using two-sided, one-sample, Student's t-test. Results indicate a significantly positive association between mood ratings and choice residuals (β = 0.05 ± 0.02, t[29] = 2.32, and p=0.027,), meaning that variability in choices, beyond that induced by the three dimensions of the offer, was indeed partially explained by mood fluctuations.

## Modeling mood fluctuations

Because frequent assessment of subjective emotional states can lead to inconsistent ratings (**Kahneman et al., 2004**), mood ratings were collected in a minority of the trials (25%). To retrieve mood levels in trials where no rating was provided, we used computational modeling. To ensure the model was based on solid grounds, we checked beforehand that mood ratings were influenced by the expected factors related to the quiz task.

We first verified that our manipulation of feedbacks was effective (**Figure 1d**): indeed the proportion of correct feedbacks was higher in episodes with intended high vs. low correct feedback rate (0.60±0.01 vs. 0.21% ± 0.01%; t[29] = 22.99, p=4.10$^{-20}$; two-sided, paired Student's t-test) and so was mood rating (0.22±0.04 to –0.12±0.06; t[29] = 3.76, p=8.10$^{-4}$, two-sided, paired, Student's t-test).

We then fitted individual mood ratings with a previously established computational model (**Vinckier et al., 2018**) that formalizes a reciprocal influence between mood and feedback (see Materials and methods). This means that mood level increases with positive feedback (reciprocally, decreases with negative feedback) and that, in turn, feedbacks are perceived as more positive when mood is higher (reciprocally, more negative when mood is lower, see **Figure 1—figure supplement 1**).

Posterior estimates of free parameters indicated that indeed, the weight of feedback on mood (parameter $\omega_f$) was significantly positive across subjects (t[29] = 4.72 and p=6.10$^{-5}$; two-sided, one-sample, Student's t-test). Conversely, the weight of mood on feedback (parameter δ) was also significantly positive (t[29] = 3.30 and p=0.0026). The forgetting factor was 0.69±0.04, meaning that feedback received five trials earlier still had an impact corresponding to 23% of the most recent feedback. In addition, the model included a linear effect of time, weighted by parameter $\omega_t$, which was not significantly different from zero, meaning that task duration had no significant influence on mood. Using fitted parameters for every subject, a TML was then estimated on a trial-by-trial basis. As seen with mood rating (**Figure 1d**), TML was significantly higher during episodes with high vs. low correct feedback rate (t[29] = 4.19 and p=2.10$^{-4}$; two-sided, paired, Student's t-test). At the group level, Bayesian comparison indicated that the mood model was far more plausible (exceedance probability Xp = 1) than the null model assuming no influence of feedback but only the effect of time. However, the results of model comparison were more variable at the individual level (see **Figure 1d**), with the null model doing better than the mood model in 6 subjects (out of 30).

## Intracerebral activity underpinning mood fluctuations

The next step was to establish a link between mood level and iEEG signals. The iEEG dataset included a total of 3494 recording sites (bipolar montage, see Materials and methods) acquired from 30 patients (**Figure 2—figure supplement 1**). For each subject, recording sites were localized within the native anatomical brain scan and labeled according to either MarsAtlas (**Auzias et al., 2016**), **Destrieux et al., 2010**, or **Fischl et al., 2002** parcellation schemes, with slight modifications (see Materials and methods). Out of the 3494 recording sites, 3188 were free from artifacts and located within the gray matter of the 50 covered areas. Among these areas, 42 regions (n=3154 sites) with at least 10 recording sites across at least 4 subjects were retained in the electrophysiological analyses (see Materials and methods). We initially focused on broadband gamma activity (50–150 Hz, BGA), as converging lines of evidence showed that BGA correlates positively with both fMRI and spiking activity (**Lachaux et al., 2007**; **Manning et al., 2009**; **Mukamel et al., 2005**; **Niessing et al., 2005**; **Nir et al., 2007**; **Winawer et al., 2013**) but subsequently took all frequency bands into consideration.

We extracted BGA time series from all retained recording sites and performed multiple linear regressions (one per time point) across trials between BGA and mood rating (i.e. using only 25% of trials with ratings) or TML (i.e. using all trials unless specified otherwise, see below). We time-locked

this analysis to the (−4 to 0 s) time window before prospects onset, i.e., during the time period corresponding to the rest or mood assessment period. Significance was tested across all recording sites within each region of interest (ROI), using two-sided, one-sample Student's t-tests on regression estimates, with correction for multiple comparisons across time points using a non-parametric cluster-level statistic (see Materials and methods). To identify brain regions that were reliably associated with mood fluctuations (*Figure 2a*), we focused our analyses on ROIs within which BGA was significantly associated with both mood rating and TML, with the additional constraint that the sign of the correlation had to be consistent.

The vmPFC (n=91 sites from 20 subjects) was the only ROI for which we found a positive correlation (*Figure 2b*; *Source data 1*; Table S2 in *Supplementary file 1*) between BGA and both mood rating (best cluster: −1.37 to −1.04 s, sum[t(90)]=122.3, and $p_{corr}$=0.010) and TML (best cluster: −0.57 to −0.13 s, sum[t(90)]=132.4, and $p_{corr}$=8.10$^{-3}$). Conversely, we found a negative correlation in a larger brain network encompassing the dorsal aIns (daIns) (n=86 sites from 28 subjects, *Figure 2b*; *Source data 1*; Table S2 in *Supplementary file 1*), in which BGA was negatively associated with both mood rating (best cluster: −3.36 to −2.51 s, sum[t(85)]=−325.8, and $p_{corr}$<1.7.10$^{-5}$) and TML (best cluster: −3.13 to −2.72 s, sum[t(85)]=−136.4, and $p_{corr}$=9.10$^{-3}$). The additional brain regions in which BGA was negatively associated with mood level (indexed with either mood rating or TML) involved the superior portion of the dorsolateral prefrontal cortex, the lateral and rostro-mesial visual cortex, the ventral motor cortex, and the dorsomedial premotor cortex (Table S2 in *Supplementary file 1*). Note that in the ventral aIns (vaIns), BGA was negatively associated with mood rating, but not with TML.

To avoid making the implicit assumption, related to fixed-effect analyses across recording sites, that differences between brains are negligible, we checked that these effects remained significant using generalized linear mixed-effects models, which separately account for inter-site and inter-individual variance (Tables S2-3 in *Supplementary file 1*).

Thus, our two a priori ROIs signaled mood level with opposite signs, whether the correlation with BGA was tested during mood rating only (i.e. using only 25% of trials with ratings) or extended to all trials using TML to extrapolate mood ratings. We also verified that the association between TML and BGA (averaged within the best temporal cluster for each ROI) remained significant when only considering the 75% of trials with no rating (vmPFC: β=0.031 ± 9.10$^{-3}$, t[90] = 3.59, and p=5.10$^{-4}$; daIns: β=−0.02 ± 8.10$^{-3}$, t[85] = −2.67, and p=9.10$^{-3}$).

To assess whether vmPFC and daIns regions would represent separate components of mood, we entered them into a single regression model meant to explain TML. In order to obtain the time course of mood expression in the two ROIs (*Figure 2c*), we performed regressions between TML and BGA from all possible pairs of vmPFC and daIns recording sites recorded in a same subject (n=247 pairs of recording sites from 18 subjects, see Materials and methods) and tested the regression estimates across pairs within each ROI for each time point. The regression estimates were significant predictors of TML in both regions, but with opposite signs (best cluster for the vmPFC: −0.71 to 0 s, sum[t(246)]=414.1, and $p_{corr}$<1.7.10$^{-5}$; best cluster for daIns: −3.05 to −2.76 s, sum[t(246)]=−113.8, $p_{corr}$=0.018).

Thus, BGA in the two main ROIs carried non-redundant information about mood level. To further investigate which component of mood was signaled by each region (*Figure 2d*), we regressed BGA against TML separately for high- and low-mood trials (35% with highest vs. lowest TML). In the vmPFC, regression estimates were significantly positive for high-mood trials only (β$_{high\ TML}$ = 0.17 ± 0.08, t[90] = 2.10, and p=0.039; two-sided, one-sample, Student's t-test), not for low-mood trials. Conversely, in the daIns, regression estimates only reached significance for low-mood trials (β$_{low\ TML}$ = −0.14 ± 0.06, t[85] = −2.39, and p=0.019), not high-mood trials. This double dissociation suggests that the vmPFC signals mood level when it gets better than average and the daIns when it gets worse than average.

In order to check that TML was expressed above and beyond the component related to the last feedback (*Figure 2e*), we splitted TML into two regressors, one representing the last feedback only (+1 or −1) and one representing the integration of all preceding feedbacks, which is equivalent to TML estimated at the previous trial (see Materials and methods). BGA in the best time clusters identified above was still significant predictors of TML (after removing the influence of the last feedback) in both the vmPFC (β$_{[−0.57, −0.13]}$=0.024 ± 8.10$^{-3}$, t[90] = 2.99, and p=4.10$^{-3}$) and daIns (β$_{[−3.13, −2.72]}$=−0.021 ± 8.10$^{-3}$, t[85] = −2.64, and p=0.010; two-sided, one-sample, Student's t-test).

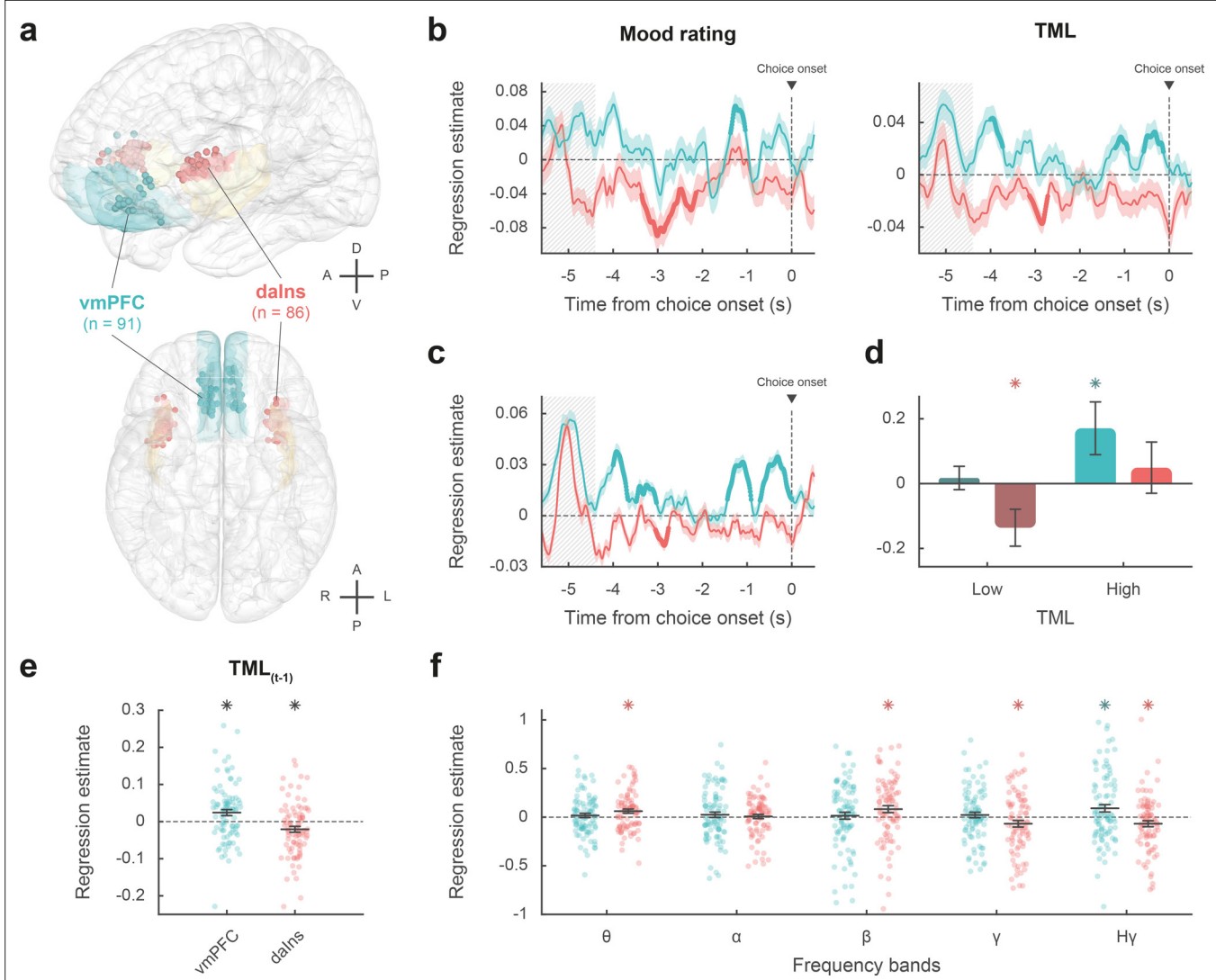

**Figure 2.** Intracerebral activity underpinning mood fluctuations. (**a**) Anatomical location of the ventromedial prefrontal cortex (vmPFC) (blue) and dorsal anterior insula (daIns) (red) in the standard Montreal Neurological Institute template brain, along with all recording sites located in these areas (dots) and the entire insula (in yellow). Anterior (A), posterior (P), dorsal (D), ventral (V), left (L), and right (R) directions are indicated. (**b**) Time course of estimates obtained from the regression of broadband gamma activity (BGA) against mood rating (left: using the 25% of trials with mood ratings) or theoretical mood level (TML) (right: using all trials – with and without ratings). (**c**) Time course of estimates obtained from the regression of vmPFC and daIns BGA included in a same general linear model (GLM) fitted to TML (using all trials). In panels b–c, lines indicate means and shaded areas ± SEM across recording sites. Bold lines indicate significant clusters ($p_{corr}$<0.05). Gray hatched areas indicate the time window within which the quiz feedback was provided to subjects. (**d**) Average estimates (within the best temporal cluster for each region of interest [ROI]) obtained from the regression of BGA against the 35% lower or higher TML. Bars are means and error bars are SEM across recording sites (vmPFC: n = 91; daIns: n = 86). (**e**) Average estimates (over the baseline window: –4 to 0 s before choice onset) obtained from the regression of BGA against TML after excluding the last feedback. Dots represent individual regression estimates for each recording site and horizontal lines and error bars respectively represent mean and SEM across sites within each ROI. (**f**) Association between TML and activity in frequency bands. For each frequency band, power time series were averaged over the baseline window and entered in a regression model meant to explain TML. θ : 4–8 Hz; α: 8–13 Hz; β: 13–33 Hz; γ: 33–49 Hz; Hγ: 50–150 Hz. In panels d–f, stars indicate significance (p<0.05) of regression estimates (two-sided, one-sample, Student's t-test).

The online version of this article includes the following figure supplement(s) for figure 2:

**Figure supplement 1.** Anatomical location of all recording sites (n=3188 sites acquired from 30 epileptic patients) in the standard Montreal Neurological Institute template brain.

Finally, to ensure that our a priori focus on BGA was justified, we explored activity in other frequency bands (*Figure 2f*). For each frequency band and recording site, power time series were averaged over the pre-choice time window (−4 to 0 s) and regressed against TML. In the vmPFC, regression estimates were only significant in the high-gamma band ($\beta_{H\gamma}$=0.091 ± 0.04, t[90] = 2.41, and p=0.018; two-sided, one-sample, Student's t-test). In the daIns however, regression estimates were not only significantly negative in the gamma and high-gamma bands ($\beta_{H\gamma}$=−0.067 ± 0.03, t[85] = −2.18, and p=0.032; $\beta_\gamma$ = −0.067 ± 0.03, t[85] = −2.0, and p=0.048) but also significantly positive in the theta and beta bands ($\beta_\beta$=0.082 ± 0.04, t[85] = 2.31, and p=0.023; $\beta_\theta$=0.062 ± 0.02, t[85] = 2.85, and p=6.10$^{-3}$). To check whether low-frequency bands could provide additional information about mood level, we fitted TML with all possible general linear models (GLMs) containing BGA together with any combination of other frequency bands (see Materials and methods). In both vmPFC and daIns, Bayesian model selection designated the BGA-only GLM as providing the best account of TML (vmPFC: expected frequency Ef = 0.99 and exceedance probability Xp = 1; daIns: Ef = 0.99 and Xp = 1). Thus, even if other frequency band activity was significantly related to TML in daIns, it carried redundant information relative to that extracted from BGA. Hence, BGA was the best neural proxy for mood level, at least in our two main ROIs.

## Impact of baseline intracerebral activity on decision making

To identify which mood-related regions impacted choices, we regressed across trials the residuals of choice model fit against BGA (time-locked to choice onset) for every time point and recording site.

The brain regions showing a significant positive association between choice and BGA included the mid-cingulate, dorsal, rostral, and ventral medial prefrontal cortex, the vaIns, the rostral inferior temporal cortex, and parts of the visual and motor cortex. Negative association between choice and BGA was observed in regions such as the rostro-ventral premotor cortex, the posterior and daIns, and the ventral inferior and the superior parietal cortex (Table S3 in *Supplementary file 1*). Therefore, among regions that encoded mood levels in their baseline, regression estimates were significant in only two ROIs (*Figure 3a*; *Source data 2*), namely a positive correlation in vmPFC (best cluster: −1.64 to −1.31 s, sum[t(90)]=91.2, and p$_{corr}$=0.020) and negative correlation in daIns (best cluster: −0.95 to −0.67 s, sum[t(85)]=−85.2, and p$_{corr}$=0.029).

Taken together, these results mean that vmPFC and daIns baseline BGA not only express mood in opposite fashion but also had opposite influence on upcoming choice. To clarify which trials contributed to the significant association between choice and BGA, we separately regressed the residuals of choice model fit against BGA across either high- or low-mood trials (median split on TML; *Figure 3b*). In the vmPFC, regression estimates were significantly positive for high-mood trials only ($\beta_{high\,TML}$ = 0.06 ± 0.01, t[90] = 5.64, and p=2.10$^{-7}$; two-sided, one-sample, Student's t-test), not for low-mood trials. Conversely, in the daIns, regression estimates only reached significance for low-mood trials ($\beta_{low\,TML}$ = −0.05 ± 0.01, t[85] = −4.63, and p=1.10$^{-5}$), not for high-mood trials. This double dissociation suggests that the vmPFC positively predicts choice when mood gets better than average, and the daIns negatively predicts choice when mood gets worse than average.

Next, we tested whether baseline activity in our two ROIs was carried over choice-related activity so as to bias decision making (*Figure 3c*). For each contact of a given ROI, we regressed choices against BGA separately for trials with high and low baseline BGA (in the time window identified in the preceding analysis for each ROI). For the vmPFC, we found that when baseline BGA was high, choice-related BGA positively predicted choices (best cluster: −1.41 to −1.16 s, sum[t(90)]=70.4, and p$_{corr}$=0.035). In contrast, for the daIns, when baseline BGA was high, choice-related BGA negatively predicts choices (best cluster: −0.74 to −0.48 s, sum[t(85)]=−78.2, and p$_{corr}$=0.022).

Finally, we investigated the computational mechanism through which baseline BGA (in the time window identified above for each ROI) could modulate decision making. For each contact of a given ROI, we fitted choices with a model in which the weights on the different terms of expected utility ($k_0$, $k_g$, $k_l$, and σ; see Materials and methods) were modulated by baseline BGA (*Figure 3d*). Comparing posterior parameters obtained with high- vs. low-BGA trials (obtained by median split), we found that $k_g$ was significantly enhanced by vmPFC baseline (t[90] = 3.16 and p=2.10$^{-3}$; two-sided, paired, Student's t-test), whereas $k_l$ was significantly enhanced by daIns baseline (t[85] = 2.90 and p=5.10$^{-3}$). This suggests that iEEG baseline fluctuations in these two mood-related regions had opposite effects:

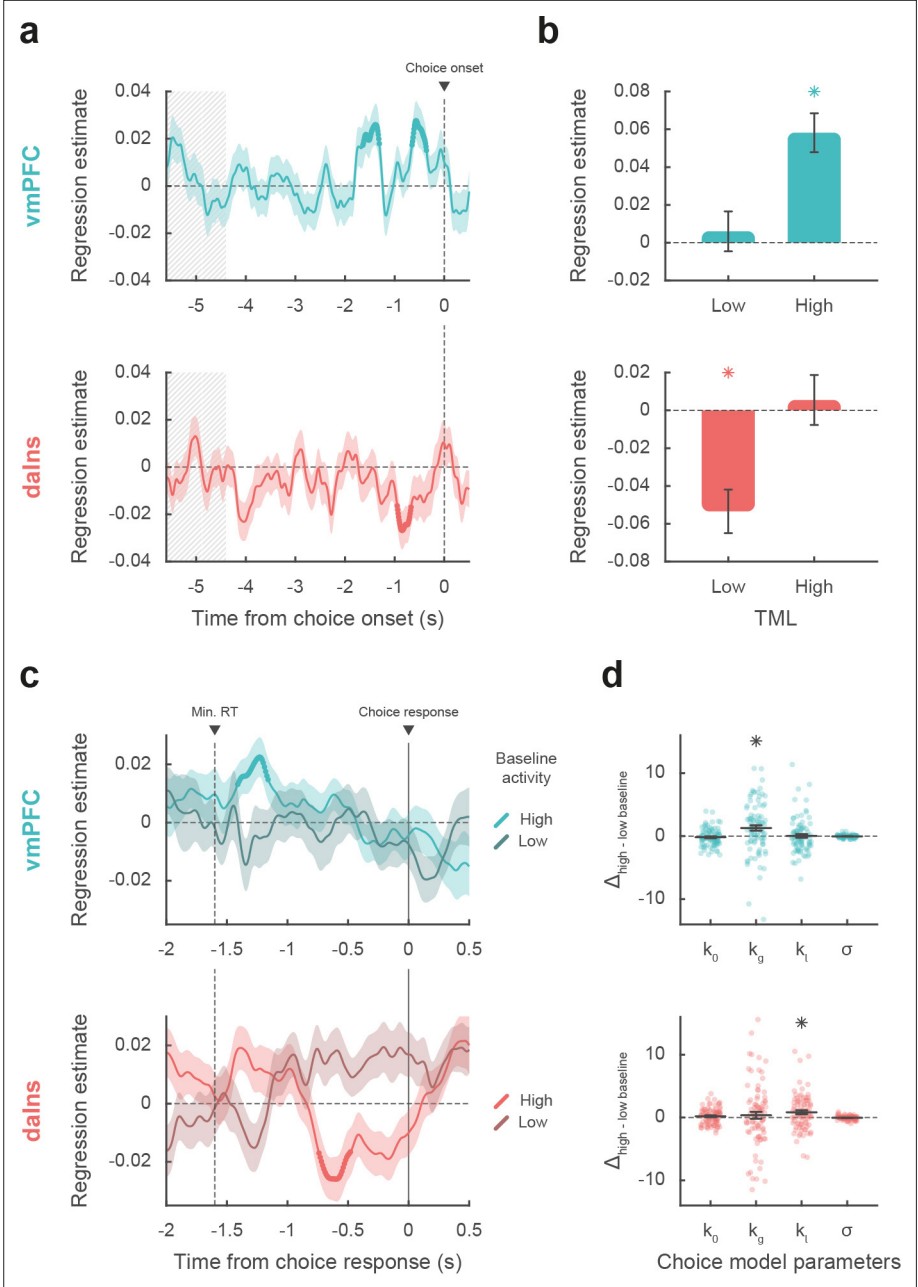

**Figure 3.** Impact of intracerebral activity on decision making. (**a**) Intracerebral electroencephalographic (iEEG) activity predicting choice. Plots show the time course of estimates obtained from the regression of choice residuals against broadband gamma activity (BGA) in ventromedial prefrontal cortex (vmPFC) (top, blue) and dorsal anterior insula (daIns) (bottom, red), averaged across recording sites ± SEM (shaded areas). Bold lines indicate significant clusters ($p_{corr}<0.05$). Gray hatched areas highlight the time window within which quiz feedback was provided to subjects. (**b**) Average estimates obtained from regression of choice residuals against BGA, separately in trials where theoretical mood level (TML) was low or high according to a median split. Bars are means and error bars are SEM across recording sites (vmPFC: n = 91; daIns: n = 86). (**c**) Time course of estimates from the regression of choices against BGA performed separately for high vs. low baseline BGA trials in vmPFC (top, blue) and daIns (bottom, red). Dashed line indicates the minimum response time of trials used in the regression (1.6 s). (**d**) Impact of baseline BGA on choice model parameters. Plots show the difference in model weights (posterior parameters) between fits to high- vs. low-BGA trials. Significant modulation was only found for $k_g$ (weight on potential gain) with vmPFC BGA and for $k_l$ (weight on potential loss) with daIns BGA. Dots represent individual data and horizontal lines and error bars respectively represent mean and SEM across recording sites. Stars indicate significance ($p<0.05$; two-sided, one-sample, Student's t-test).

increased vmPFC BGA would lead to overestimating gain prospects, while increased daIns BGA would lead to overestimating loss prospects.

We also verified that the main findings of this study remained significant (or borderline) when using group-level random-effects analyses (Table S4 in *Supplementary file 1*, see Materials and methods), even if this approach is underpowered and unadjusted for sampling bias (some subjects having very few recording sites in the relevant ROI).

## Discussion

In the present study, we used a large dataset of iEEG signals recorded in 30 subjects to provide a neuro-computational account of how mood fluctuations arise and impact risky choice. We found that (1) baseline BGA in vmPFC and daIns was respectively signaling periods of high and low mood induced by incidental feedbacks, (2) beyond BGA, oscillatory activity in other frequency bands did not provide any additional information about mood level, (3) high vmPFC baseline BGA promoted risk taking by selectively increasing the weight of potential gains, whereas high daIns baseline BGA tempered risk taking by increasing the weight of potential losses, and (4) baseline BGA in both regions was carried over BGA during decision making, which was predictive of the eventual choice. In the following, we successively discuss how BGA in these two regions relate to mood and choice.

### BGA and mood

To avoid asking subjects to rate their mood on every trial, we used a computational model to interpolate mood level between mood ratings. The model was inspired from previous suggestions (*Eldar and Niv, 2015*; *Rutledge et al., 2014*) and already validated in a previous publication (*Vinckier et al., 2018*). The basic assumption is that mood is nothing but a weighted sum of positive and negative events (here, feedbacks received in the quiz task), the more recent having more weight than the more distant in the past. In addition, the model postulates that the way feedbacks are perceived are also affected by mood, in the sense that a same feedback is perceived as more positive when we are in a better mood. This makes reciprocal the influence between internal states (mood) and external events (feedback). The model was inverted on the basis of mood ratings and compared to a null model, where mood is just drifting with time, without any influence of feedback. Even if the mood model was largely winning the Bayesian comparison at the group level, its evidence was surpassing the null model in about two-thirds of subjects, a proportion close to that reported in the previous publication (*Vinckier et al., 2018*). We nevertheless kept all subjects in subsequent analyses to reach more robust conclusions that might be generalized to the entire population. Posterior estimates of free parameters confirmed the reciprocal influence between feedback and mood, the forgetting factor suggesting that mood was still impacted by the feedback received five trials in the past (about 2 min before), with a weight about four times lesser than the last feedback. Thus, the time scale of mood fluctuations was longer than acute emotional reactions to single stimuli but much shorter than those observed in mood disorders.

When looking for neural correlates of mood fluctuations, we used both raw ratings and modeled mood levels as probes. The results were qualitatively similar but statistically more significant with TML because it allowed including all trials in the analysis. To identify brain regions that could mediate the impact of mood on choice, we selected all regions that were significantly associated to both mood level (good or bad) and choice (safe or risky). We found the two main bilateral regions that were already identified with fMRI (*Vinckier et al., 2018*), plus unilateral visual or motor regions that were likely related to the side of the behavioral response. We note however that the aIns region defined anatomically corresponds to a dorsal part of the cluster identified with fMRI (hence the appellation of daIns) and that the vmPFC region included many recorded sites located in more ventral areas than the fMRI cluster. As with fMRI, we observed that correlation with mood level was positive in the vmPFC and negative in the daIns. This is consistent with many studies implicating the vmPFC in reward learning and the daIns in punishment learning (*Fouragnan et al., 2018*; *Garrison et al., 2013*; *Liu et al., 2011*; *Palminteri and Pessiglione, 2017*), including studies using iEEG (*Gueguen et al., 2021*). What we additionally show here is that the vmPFC and daIns are not just sensitive to the last reward or punishment outcome as in learning paradigms but integrate feedbacks over a longer time scale. Indeed, the time course of regression estimates showed significant association with mood in

time windows much later than the response to the last feedback, and the association remained significant when removing the impact of the last feedback from the computation of TML used as regressor. We note that some expected regions are missing in our list, notably the ventral striatum, which has been shown to correlate positively with mood level (*Eldar and Niv, 2015*; *Rutledge et al., 2014*; *Young and Nusslock, 2016*). This region could not be investigated here because it was simply not sampled by the electrodes implanted for clinical purposes (i.e. for localization of epileptic foci). More generally, an inherent limitation to iEEG is that parts of the brain are less sampled than others, so null results should be considered with caution, because of huge differences in statistical power across regions. Nevertheless, whole-brain analysis using fMRI showed that with this induction procedure, activity in other regions such as the ventral striatum or dorsal anterior cingulate cortex adds nothing to decoding of mood level (*Vinckier et al., 2018*).

Having identified the two main regions reflecting mood fluctuations, we tested whether mood fluctuations were associated to similar shifts in the frequency of oscillatory activity as was previously suggested (*Bijanzadeh et al., 2013*; *Kirkby et al., 2018*; *Rao et al., 2018*). On the contrary, we observed that higher mood had opposite effects on high-frequency oscillations in the two regions, with increased BGA in the vmPFC but decreased BGA in the daIns. Thus, mood fluctuations were better accounted for by relative high-frequency activity (BGA) in the two opponent regions. Indeed, when included in the same model, vmPFC and daIns BGA were both significant predictors of mood level. Conversely, including low-frequency activity in the model did not help predicting mood level. The information carried in low-frequency activity was therefore at best redundant with that of high-frequency activity, whereas the two regions carried at least partly independent information. When analyzing separately bad, neutral, and good mood levels, we found that higher BGA signaled the two extreme tertiles: good mood in the vmPFC and bad mood in the daIns. This result suggests that the two regions were specifically concerned with good and bad mood, explaining why they were not just anti-correlated, as previously suggested (*Kucyi et al., 2020*). Although we used a unidimensional rating going from bad to good mood, it could be that mood is in fact bidimensional, meaning that

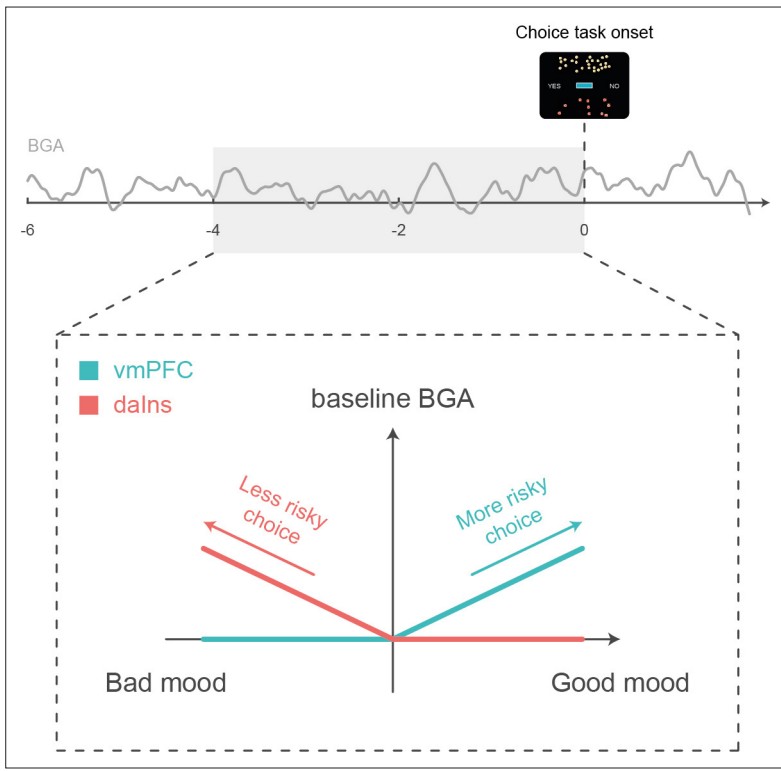

**Figure 4.** Schematic summary. A higher baseline broadband gamma activity (BGA) in the ventromedial prefrontal cortex (vmPFC) was both predicted by good mood and associated with higher accept rates in the subsequent choice task, whereas a higher baseline BGA in the dorsal anterior insula (daIns) was both predicted by bad mood and associated with higher reject rates.

good and bad mood would be better conceived as independent components (*Watson et al., 1988*), relying on distinct brain systems. This is in keeping with clinical practice: absence of positive affect or excess of negative affect is considered as two independent criteria for depression.

The observation that mood is signaled by distinct brain systems and not different frequency bands all over the brain as in previous iEEG studies (*Bijanzadeh et al., 2013*; *Kirkby et al., 2018*; *Rao et al., 2018*) might be related to the use of positive and negative events to induce mood fluctuations and not direct stimulation of the cortical surface, which might put constraints on oscillatory activity. Besides, the good correspondence with previous fMRI results validates the shared view that BGA represents a time-resolved neural index of local neuronal activity, which typically correlates with both hemodynamic response and local spiking activity (*Lachaux et al., 2007*; *Manning et al., 2009*; *Mukamel et al., 2005*; *Niessing et al., 2005*; *Nir et al., 2007*; *Winawer et al., 2013*). We have confirmed this correspondence every time we compared iEEG and fMRI activity during the same behavioral paradigm (*Gueguen et al., 2021*; *Lopez-Persem et al., 2020*).

## BGA and risky choice

Choice to accept or reject the challenge in our task was significantly modulated by the three attributes displayed on screen: gain prospect (in case of success), loss prospect (in case of failure), and difficulty of the challenge. We combined the three attributes using a standard expected utility model and examined the residuals after removing the variance explained by the model. Those residuals were significantly impacted by mood level, meaning that on top of the other factors, good/bad mood inclined subjects to accept/reject the challenge. The same was true for neural correlates of mood: higher baseline BGA in the vmPFC/daIns was both predicted by good/bad mood and associated to higher accept/reject rates, relative to predictions of the choice model. Thus, different mood levels might translate into different brain states that predispose subjects to make risky or safe decisions (*Figure 4*).

The remaining question is how baseline BGA in these regions can be mechanistically related to risky choice. To address this question, we examined BGA during decision making, which could be separated from baseline BGA thanks to the temporal resolution of iEEG recordings. We found some carry-over: higher BGA in the vmPFC/daIns again predicted the likelihood of accept/reject choices. This does not mean that the link between BGA and choice was maintained all along; it was actually lost at the offer onset and retrieved before the behavioral response. This pattern could not be observed with fMRI but was already reported regarding the subjective value signals (*Lopez-Persem et al., 2020*). It suggests that the impact of baseline activity in mood-related regions was mediated by the contribution of the same regions (vmPFC and daIns) to the decision-making process. These regions have already been implicated in computing the subjective values that are compared to make a choice, with the vmPFC and daIns, respectively, providing positive and negative value signals (*Bartra et al., 2013*; *Pessiglione and Delgado, 2015*). Thus, the relative activity of two regions may provide an impulse to accept or reject an offer, depending on whether the differential is positive or negative.

An additional explanation would be that the two regions not only provide subjective values with opposite signs but also that they provide values that differ in how the different attributes are weighted. To test this idea, we compared parameters of the choice model fitted separately onto trials with high vs. low baseline BGA in the vmPFC and daIns. We found that high BGA in the vmPFC/daIns was respectively associated to overweighting of gain and loss prospects and not just the constant that captures the global tendency to accept or reject. This result makes the link with the idea that we may see a glass half-full or half-empty when we are in a good or bad mood, possibly because we pay more attention to positive or negative aspects. It is also consistent with reports that pre-stimulus spiking or hemodynamic activity in the vmPFC predicts how much rewards are liked (*Abitbol et al., 2015*; *Lopez-Persem et al., 2020*), whereas pre-stimulus activity in the daIns predicts how aversive punishments are perceived (*Caria et al., 2010*; *Wiech et al., 2010*). Overall, our findings suggest that mood-related neural fluctuations impact the valuation process in a valence-specific way, implemented by functionally opponent brain systems.

An obvious limitation of these findings is that although the dataset was collected in patients with epilepsy, we make interpretations as if it was collected in healthy subjects, assuming that pathological activity did not influence the neural or cognitive processes of interest. What was reassuring is that the behavior was remarkably comparable to that of young subjects performing a similar task, while the

iEEG results were fully compatible with fMRI results (*Vinckier et al., 2018*). Also, it does not seem very plausible that artifacts would by chance correlate, across task trials in patients with different forms of epilepsy, with the computational variables that were related to both neural and behavioral results. Indeed, in previous studies, where we compared results with and without removing artifacted trials, results were unchanged. In fact, those trials can only degrade the correlation with computational variables, such that keeping them ensures that results would be more robust to replication.

In summary, we used intracerebral recordings in humans to specify the neuro-computational mechanisms through which mood fluctuations might arise from external events and impact risky choice. At longer time scales, these mechanisms could explain why people take more gambles (or less) after incidental events such as the victory (or defeat) of their favorite sport team, an effect that might be exacerbated in pathological conditions such as during manic (or depressive) episodes. The impact of mood on choice is a form of generalization, across different sources of reward and punishment, that may be catastrophic in pathological cases, leading people to believe that because they experienced positive (or negative) outcome in some part of their life, they are likely to succeed (or fail) in any other.

## Materials and methods

### Patients and electrode implantation

Intracerebral recordings were obtained from 30 patients suffering from drug-resistant focal epilepsy (39.5±1.9 years old, 14 females, see demographic details in Table S1 in *Supplementary file 1*) in 7 different epilepsy centers (Rennes University Hospital: n=10; Grenoble University Hospital: n=9; Lyon Neurological Hospital: n=3; Prague Motol University Hospital: n=3; Marseille La Timone Hospital: n=2; Toulouse University Hospital: n=2; Nancy University Hospital: n=1). These patients underwent intracerebral recordings by means of stereotactically implanted multilead depth electrodes (Stereo-electroencephalography or sEEG) in order to locate epileptic foci that could not be identified by non-invasive methods. Electrode implantation was performed according to routine clinical procedures, and all target structures for the pre-surgical evaluation were selected strictly according to clinical considerations with no reference to the current study. 9–20 semi-rigid electrodes were implanted per patient. Each electrode had a diameter of 0.8 mm and, depending on the target structure, contained 6–18 contact leads of 2 mm wide and 1.5–4.1 mm apart (Dixi Medical, Besançon, France). All patients gave written, informed consent before their inclusion in the present study, which received approval from the local ethics committees (CPP 09-CHUG-12, study 0907; CPP18-001b/ 2017-A03248-45; IRB00003888; CER No. 47–0913).

### iEEG recordings

Neuronal recordings were performed using video-EEG monitoring systems that allowed for simultaneous recording of 128–256 depth-EEG channels sampled at 512, 1024, or 2048 Hz (depending on the epilepsy center). Acquisitions were made with Micromed (Treviso, Italy) system and online band-pass filtering from 0.1 to 200 Hz in all centers, excepting for Prague (two Natus systems were used: either a NicoleteOne with a 0.16–134 Hz band-pass filtering or a Quantum NeuroWorks with a 0.01–682 Hz band-pass filtering) and Marseille (Deltamed system, 0.16 Hz high-pass filtering). Data were acquired using a referential montage with reference electrode chosen in the white matter. Before analysis, all signals were re-referenced to their nearest neighbor on the same electrode, yielding a bipolar montage.

### Behavioral tasks

Presentation of visual stimuli and acquisition of behavioral data were performed on a personal computer using custom Matlab scripts implementing the PsychToolBox libraries (*Brainard, 1997*). All subject responses were done with a gamepad (Logitech F310S) using both hands. Subjects performed a choice task combined with a mood induction procedure adapted from a previous study (*Vinckier et al., 2018*). Subjects completed two (n=6) or three (n=24) sessions of the experiment, consisting of 64 trials each, for a total of 128 or 192 trials. Each trial included three sub-parts: a quiz task, a rest or mood assessment period, and a choice task.

## Quiz task

In this task, a question and four possible answers were displayed on the screen. The question was randomly selected from a set of 256 possible questions that were adapted from the French version of the 'trivial pursuit' game (e.g. where is Park Güell located?) which were slightly adapted for Prague's epilepsy center. Subjects had to select the correct answer using the up and down keys and confirm their answer using the confirmation key within a maximum of 8 s. A feedback of 1 s was finally given (either a smiling face with a bell sound or a grimacing face with a buzzer sound) immediately after the answer or at the end of the available time if no answer had been made.

In order to predictably manipulate subjects' mood, episodes of high and low correct response rates were created unbeknownst to them by handling questions difficulty and feedbacks. Thus, questions were sorted by difficulty (assessed by mean accuracy estimated previously in a sample of healthy subjects; *Vinckier et al., 2018*) and grouped so as to create easy and hard episodes. Within a given session of 64 trials, we created one episode of 20 trials with easy questions followed by one episode of 20 trials with hard questions and three episodes of 8 trials with questions of medium difficulty. The episodes were organized so that easy and hard episodes were always preceded and followed by episodes of medium difficulty (e.g. medium – easy – medium – hard – medium). The order of easy and difficult episodes was counterbalanced across sessions and subjects. Furthermore, feedback was biased such that a wrong answer could lead to a positive feedback (whereas a correct answer always led to a positive feedback). The proportion of biased feedback depended on the difficulty of the episode: 50% in easy, 25% in medium, and 0% in hard episode. Post-hoc debriefing showed that no subject was aware of this manipulation.

## Mood rating/rest period

Mood assessment or rest period began with a 500±100 ms black screen used to ensure that a reasonable delay occurred after the last quiz feedback (which lasted 1 s). In 25% of the trials (i.e. in 16 out of 64 trials), the quiz task was followed by a mood rating task in which subjects were explicitly asked to rate their mood by answering the following question: 'how do you feel right now?'. Subjects had to answer by moving a cursor from left (very bad) to right (very good) along a continuous visual analog scale (100 steps) with left and right hand response buttons. Subjects were given at least 4 s to confirm their ratings with the confirmation button. Their response was then maintained on the screen until the end of the 4 s so that they had no reason to speed-up their estimation. However, subjects were also discouraged from being too long to respond, as when they had not confirmed their rating within 4 s, a red message was displayed on the screen saying 'please validate your answer.' The initial position of the cursor on the scale was randomized to avoid confounding mood level with movements' quantity. The position of mood ratings across trials within a session was evenly distributed and pseudo-randomized so that mood ratings were not predictable for the subjects, with the additional constraints that two mood ratings were spaced by a minimum of two trials and a maximum of six trials. In the remaining 75% of trials, the quiz task was followed by a rest period consisting of a 4 s black screen. Therefore, the delay between the end of the quiz feedback and the beginning of the choice task was kept to a minimum of 4.5±0.1 s.

## Choice task

The choice task began with the presentation of an offer consisting of three dimensions: a gain prospect (represented by a bunch of 10-cent coins, range: 1–5 €), a loss prospect (represented by crossed out 10-cent coins, range: 1–5 €), and the upcoming challenge difficulty (represented by the size of a target window located at screen center, range: 1–5 corresponding to 75–35% theoretical success; see training section for further details about how difficulty was adjusted to each subject). Subjects were asked to accept or reject this offer by pressing a left or right button depending on where the choice option ('yes' or 'no') was displayed. Subjects' choice determined the amount of money at stake: accepting meant that they would eventually win the gain prospect or lose the loss prospect based

on their performance in the upcoming challenge, whereas declining the offer meant playing the challenge for minimal stakes (winning 10 cents or losing 10 cents).

The sequence of trials was pseudo-randomized such that all possible combinations of the three dimensions (gains, losses, and challenge difficulty), continuously sampled along four intervals ([1-2], [2–3], [3–4], and [4–5]), were displayed for one subject across sessions, with greater sampling of medium difficulty combinations ([2-3] and [3–4]) to maximize the occurrence of undetermined choices during which small ongoing fluctuations were previously shown to bias subsequent choices (*Padoa-Schioppa, 2013*). The positions of gain and loss prospects were randomly determined to be either displayed on top or bottom of the screen, and similarly, the choice options ('yes' or 'no') were randomly displayed on the left or right.

Subjects had a free time delay to accept or decline the offer. If they declined the offer, a 500 ms screen displayed the new offer (only gains and loss prospects changed so that subjects performed the challenge for a minimal stake of 10 cents). Thus, an important feature of the choice task was that the challenge was performed regardless of the choice answer to prevent subjects to eventually reject more offers to decrease experiment duration.

The challenge started 200 ms after choice confirmation: a ball appeared on the left of the screen and moved, horizontally and at constant speed, toward screen center. Subjects were asked to press the confirmation button when they thought the ball was inside the basket displayed at screen center (i.e. the target window which size index the difficulty of the challenge). To facilitate the challenge, which was performed without any feedback, the ball always reached the center of the target after 1 s. Thus, the size of the target window (i.e. the difficulty of the trial) represented the margin of error tolerated in subject response time (target: 1 s after the movement onset of the ball). The larger the basket, the greater the tolerated spatiotemporal error was to consider a trial as successful, and therefore the easier was the trial. Importantly, the moving ball could only be seen during the first 500 ms (half of the trajectory), and subjects had to extrapolate the last 500 ms portion of the ball's trajectory to assess whether the ball was inside the target. No feedback was given to subjects about their performance or payoff after the challenge to prevent learning effects and also choice feedback effects on mood. However, to improve subjects' motivation to perform the task as accurately as possible, the total amount of money earned by the subjects during a session (calculated by adding gains and losses across all trials) was displayed at the end of a session.

## Training

Before the main experiment, a training – divided into three steps – familiarized subjects with all subparts of the task. In the first step, subjects were familiarized with the challenge by performing 28 trials with a tolerated margin of error from ±130 to ±80 ms (56 trials if the accuracy of the first 28 trials was below 50%). Each training trial was followed by feedback informing whether the challenge was successful ('ok' in green) or missed ('too slow' or 'too fast' in red). In the second step, subjects performed 64 trials of the full choice (i.e. the challenge was always preceded by an offer), and a feedback on the money won/lost in the trial was displayed at the end of each trial. The goal was to train subjects to properly integrate the three dimensions of the offer (gains, losses, and difficulty) when making their choice. To help subjects learning the correspondence between the target size and challenge difficulty, trials were displayed by increasing difficulty level. Finally, the third and last part of the training (10 trials including 2 mood ratings) were totally similar to the main task to allow subjects to familiarize with the quiz task and the mood ratings.

Another purpose of the training was also to tailor the difficulty of the challenge to each subject's ability. To do so, a tolerated margin of error was computed for each difficulty level, ranging from 75 (level 1) to 35% (level 5) of theoretical success which we estimated from each individual subjects by assuming that errors were normally distributed. Note that during training, the difficulty levels were updated after each trial (average and SD of challenge performance were updated), while in the main task, the mean and SD of subject performance (and therefore difficulty levels) were set based on every challenge performed during the training. The range of tolerated margins of error between subjects ranged from (±72 ms [level 1] to ±22 ms [level 5]) in the most precise subject to (±198 ms to ±123 ms) in less precise one.

## Computational models

Mood ratings and choices were fitted using published computational framework (*Vinckier et al., 2018*).

### Mood model

As mood was sampled in 25% of the trials, we linearly interpolated ratings in order to get one data point per trial. In all analyses, mood ratings were z-scored. A TML was computed for each trial through the integration of quiz feedback as follows:

$$TML\left(t\right) = \omega_0 + \omega_f \sum_{j=1}^{t} \gamma^{t-j} F\left(j\right) + \omega_t t$$

where $t$ is the trial index, $\gamma$ and all $\omega$ are free parameters ($\omega_0$ is a constant and all other $\omega$ are weights on the different components; $\gamma$, with $0 \leq \gamma \leq 1$, is a forgetting factor that adjusts the influence of recent events relative to older ones), and $F$ is the subjective perception of feedback. This feedback was subjective as its perception was in return biased by TML:

$$F\left(t\right) = Feedback\left(t\right) + \delta \times TML\left(t-1\right)$$

where $\delta$ is a free parameter and $TML\left(t-1\right)$ is the TML carried from previous trial, before updating based on the feedback received in the current trial. The model assumes that mood effect is additive so that good mood leads events to be seen as more positive than they objectively are (a multiplicative effect would imply that a negative feedback is perceived as even worst when one is in a good mood) and allowed an asymmetrical influence of positive and negative events on mood, using a free parameter R (with R>0) for positive feedback instead of 1:

$$Feedback\left(t\right) = R \vee -1$$

This model was compared to a control model in which only time was taken into account (linear function of trial index).

### Choice model

Acceptance probability was calculated as a sigmoid function (softmax) of expected utility:

$$p\left(accept, t\right) = \frac{1}{1+e^{-\left(utility+k_t \times t\right)}}$$

where $k_t$ is a free parameter that accounts for a linear drift with time (trial index $t$) in order to capture fatigue effects. The utility function is based on expected utility theory where potential gains and losses are multiplied by probability of success ($p_s$) vs. failure ($1 - p_s$):

$$utility = k_0 + p_s \times k_g \times gain - \left(1 - p_s\right) \times k_l \times loss$$

However, distinct weights were used for the gain and loss components ($k_g$ and $k_l$, respectively), and a constant $k_0$ was added in order to capture a possible bias. The subjective probability of success ($p_s$) was inferred from the target size. The model assumes that subjects have a representation of their precision following a Gaussian assumption, meaning that the subjective distribution of their performance could be defined by its mean (the required 1 s to reach target center) and its width (i.e. SD) captured by a free parameter $\sigma$. Thus, the probability of success was the integral of this Gaussian bounded by the target window:

$$p_s = \frac{1}{\sigma\sqrt{2\pi}} \int \frac{1-}{1 + e^{\frac{-(x-1)}{2\sigma^2}}} dx$$

Both models (mood and choice models) were inverted, for each subject separately with behavioral data, using the Matlab VBA toolbox (available at https://mbb-team.github.io/VBA-toolbox/; *Rigoux,*

*2019*), which implements Variational Bayesian analysis under the Laplace approximation (*Daunizeau et al., 2014*). This algorithm not only inverts non-linear models but also estimates the model evidence, which represents a trade-off between accuracy (goodness of fit) and complexity (degrees of freedom) (*Robert, 2007*).

## Behavioral analysis

Statistical analyses were performed with Matlab Statistical Toolbox (Matlab R2018a, The MathWorks, Inc, USA). All dependent variables (raw or z-scored behavioral measures, regression estimates and model outputs) were analyzed at the subject level and tested for significance at the group level using two-sided, one or paired-sample, Student's t-tests. All means are reported along with the standard error of the mean.

## sEEG pre-processing

Before analysis, bad channels were removed based on a supervised machine-learning model trained on a learning database with channels already classified by experts and using a set of features quantifying the signal variance, spatiotemporal correlation, and non-linear properties (*Tuyisenge et al., 2018*). All signals were then re-referenced with a local bipolar montage between adjacent contacts of the same electrode to increase the spatial specificity of the effects by canceling out effects of distant sources that spread equally to both adjacent contacts through volume conduction. The average number of recording sites (one site corresponding to a bipolar contact-pair) recorded per patient was 116±37. Finally, all signals were down-sampled to 512 Hz.

## Neuroanatomy

The electrode contacts were localized and anatomically labeled using the IntrAnat Electrodes software (*Deman et al., 2018*), developed as a BrainVisa (*Rivière et al., 2009*) toolbox. Briefly, the pre-operative anatomical MRI (3D T1 contrast) and the post-operative image with the sEEG electrodes (3D T1 MRI or CT scan), obtained for each patient, were co-registered using a rigid-body transformation computed by the Statistical Parametric Mapping 12 (SPM12) (*Ashburner, 2009*) software. The gray and white matter volumes were segmented from the pre-implantation MRI using Morphologist as included in BrainVisa. The electrode contact positions were computed in the native and Montreal Neurological Institute (MNI) referential using the spatial normalization of SPM12 software. Coordinates of recording sites were then computed as the mean of the MNI coordinates of the two contacts composing the bipole. For each patient, cortical parcels were obtained for the MarsAtlas (*Auzias et al., 2016*) and Destrieux (*Destrieux et al., 2010*) anatomical atlases, while subcortical structures were generated from *Fischl et al., 2002*, as included in Freesurfer. Each electrode contact was assigned to the gray or white matter and to specific anatomical parcels by taking the most frequent voxel label in a sphere of 3 mm radius around each contact center.

The MarsAtlas parcellation scheme was mainly used to label each recording site. This atlas relies on a surface-based method using the identification of sulci and a set of 41 ROIs per hemisphere. These regions were completed with seven subcortical regions, obtained from the procedure described by Fischl et al. (as included in Freesurfer; *Deman et al., 2018*). However, based on the literature, we applied slight modifications concerning our ROIs. First, boundaries based on MNI coordinates were set to the vmPFC region so that contacts more lateral than x=±12 and more dorsal than z=10 were excluded from the parcel (*Lopez-Persem et al., 2019*). Second, MarsAtlas parcellation scheme involved the insular cortex as a single region, making it impossible to distinguish sub-insular areas that appear to have distinct functional properties in decision making (*Droutman et al., 2015*). We therefore used the Destrieux atlas (performed by Freesurfer; *Deman et al., 2018*) and MNI coordinates to segment the region corresponding to the insula into three sub-regions: (i) the vaIns corresponds to the anterior part (y<5 in MNI space) of parcels 18 (G_insular_short), 47 (S_circular_insula_ant), and 48 (S_circular_insula_inf) of the Destrieux atlas, (ii) the daIns corresponds to the anterior part (y<5 in MNI space) of parcel 49 (S_circular_insula_sup) of the Destrieux atlas, and finally (iii) the posterior insula corresponds to the posterior part (y>5 in MNI space) of parcels 17 (G_Ins_lg_and_S_cent_ins), 48, and 49 of the Destrieux atlas, leading to a total of 50 ROIs.

For statistical analyses, only the 42 ROIs with at least 10 recording sites recorded across at least four subjects were retained. Among the 3494 initial recorded sites, 3154 recording sites were located

within one of these 42 regions and were therefore kept for analysis. Note that data from both hemispheres were collapsed to improve statistical power.

## Extraction of frequency envelopes

The time course of BGA was obtained by band-pass filtering of continuous sEEG signals in multiple successive 10 Hz-wide frequency bands (e.g. 10 bands, beginning from 50 to 60 Hz up to 140–150 Hz) using a zero-phase shift non-causal finite impulse filter with 0.5 Hz roll-off. The envelope of each band-pass filtered signal was next computed using the standard Hilbert transform. For each frequency band, this envelope signal (i.e. time varying amplitude) was divided by its mean across the entire recording session and multiplied by 100. This yields instantaneous envelope values expressed in percentage (%) of the mean. Finally, the envelope signals computed for each consecutive frequency band were averaged together to provide a single time series (the broadband gamma envelope) across the entire session. By construction, the mean value of that time series across the recording session is equal to 100. Note that computing the Hilbert envelopes in 10 Hz sub-bands and normalizing them individually before averaging over the broadband interval allows to counteract a bias toward the lower frequencies of the interval induced by the 1/f drop-off in amplitude. Finally, the obtained time series were smoothed using a sliding window of 250 ms, to get rid of potential artifacts, and down-sampled at 100 Hz (i.e. one-time sample every 10 ms). BGA was normalized for each recording site by z-scoring across trials.

The envelopes of theta, alpha, beta, and gamma bands were extracted and normalized in a similar manner as the BGA except that steps were 1 Hz for theta and alpha, 5 Hz for beta, and 4 Hz for gamma. BGA frequency range was defined as 50–150 Hz, gamma as 33–49 Hz, beta as 13–33 Hz, alpha as 8–13 Hz, and theta as 4–8 Hz.

## Electrophysiological analyses

The frequency envelopes of each recording site were epoched at each trial from 5600 ms prior choice onset to 500 ms after choice onset, encompassing the quiz feedback, the rest period (or mood assessment) between quiz and choice tasks and the choice onset (display of the offer). Electrophysiological data were analyzed using GLM, providing a regression estimate for each time point and contact.

In a first GLM aimed at identifying areas underpinning mood fluctuations, power $P$ (normalized envelope) was regressed across trials against mood $M$ (real mood ratings or TML, normalized within subjects) at every time point:

$$P = \alpha + \beta M + \epsilon$$

with α corresponding to the intercept, β corresponding to the regression estimate on which statistical tests are conducted and ε corresponding to the error term.

To investigate whether vmPFC and daIns were sensitive to a specific mood state, the same GLM was performed except that BGA of each ROI was averaged over the best cluster identified in this first GLM (from –0.57 to –0.13 s before choice onset for the vmPFC and from –3.13 to –2.72 s for the daIns) and regressed across the 35% trials with the best or worst TML.

Next, to assess whether vmPFC and daIns regions would represent separate components of mood, TML was regressed across trials against BGA of these two regions at every time point as follows:

$$TML = \alpha + \beta_1 BGA_{vmPFC} + \beta_2 BGA_{daINS} + \epsilon$$

with α corresponding to the intercept, $\beta_1$ and $\beta_2$ corresponding to the regression estimates on which statistical tests are conducted and ε corresponding to the error term. For each of the 18 subjects with recording sites in both ROIs, regression was done for all possible pairs of vmPFC and daIns recording sites recorded within a given patient, leading to a total of n=247 pairs of recording sites.

To check that TML was expressed above and beyond the variance induced by the last feedback, BGA was regressed across trials $t$ against last feedback along with TML from the previous trial at every time point:

$$BGA\left(t\right) = \alpha + \beta_1 Feedback\left(t\right) + \beta_2 TML\left(t - 1\right) + \epsilon$$

where α corresponds to the intercept, $\beta_1$ and $\beta_2$ correspond to the regression estimates on which statistical tests are conducted, and ε corresponds to the error term.

Next, to assess the effect of pre-choice brain activity on acceptance rate, the residual error of choice model fit $C$ was regressed across trials against power $P$ (normalized envelope) at every time point as follows:

$$C = \alpha + \beta P + \epsilon$$

with α corresponding to the intercept, β corresponding to the regression estimate on which statistical tests are conducted and ε corresponding to the error term.

Finally, to assess how baseline activity affects choice, the frequency envelopes of each recording site were epoched at each trial from 2000 ms prior choice response to 500 ms after choice response, and trials were split into 'high' or 'low' baseline activity based on the average pre-choice BGA in the best cluster identified in the GML used to investigate effect of pre-choice brain activity on acceptance rate (from −1.64 to −1.31 s before choice onset for the vmPFC and from −0.95 to −0.67 s for the daIns). Choices were then regressed across the 40% highest or lowest baseline activity trials against BGA at every time point, as in the previous GLM. To ensure that baseline activity does not interfere with the reported effect, trials with a choice response time shorter than 1.5 up to 2 s were eliminated from this analysis. Note that the response time threshold (from 1.5 to 2 s) did not affect the result. We reported results from the analysis removing trials with a response time shorter than 1.6 s (longest response time removing less than 25% of trials).

For each ROI, a t-value was computed across all recording sites of the given ROI for each time point of the baseline window (−4 to 0 s before choice onset), independently of subject identity, using two-sided, one-sample, Student's t-tests. For all GLMs, the statistical significance of each ROI was assessed through permutation tests. First, the pairing between responses and predictors across trials was shuffled randomly 300 times for each recording site. Second, we performed 60,000 random combinations of all contacts in an ROI, drawn from the 300 shuffles calculated previously for each site. The maximal cluster-level statistics (the maximal sum of t-values over contiguous time points exceeding a significance threshold of 0.05) were extracted for each combination to compute a 'null' distribution of effect size across a time window from −4 to 0 s before choice onset (the baseline corresponding to the rest or mood assessment period). The p-value of each cluster in the original (non-shuffled) data was finally obtained by computing the proportion of clusters with higher statistics in the null distribution and reported as the 'cluster-level corrected' p-value ($p_{corr}$).

## Group-level analyses

To avoid making the implicit assumption, related to fixed-effect analyses across recording sites, that differences between brains are negligible, we checked that these effects remained significant using mixed-effects linear models, which separately account for inter-site and inter-individual variance (Tables S2-3 in *Supplementary file 1*).

Specifically, we tested in each ROI the association between mean BGA (averaged over the best temporal cluster identified in fixed-effects analyses) and (1) mood rating, (2) TML, and (3) choice with the following linear mixed-effects models:

1. BGA ~1 + mood + (1+mood | SUBJECTS) + (1+mood | RECORDING SITES:SUBJECTS).
2. BGA ~ 1 + TML + (1 + TML | SUBJECTS) + (1 + TML | RECORDING SITES:SUBJECTS).
3. Choice ~1 + BGA + (1+BGA | SUBJECTS) + (1+BGA | RECORDING SITES:SUBJECTS).

We then performed F-tests on the fixed effects estimates, with significance set at <0.05, two-tailed.

To test the association between BGA and mood, TML or choice at the group level (using random-effects analyses), we performed the same linear regression as described in the electrophysiological analyses section on BGA averaged over the best time cluster (identified by the fixed-effects approach) and across all recording sites of a given subjects located in the relevant ROI. We then conducted a two-sided, one-sample, Student's t-test on the resulting regression estimates (Table S4 in *Supplementary file 1*).

## Contribution of frequency bands

To assess the contribution of the different frequency bands to mood representation, TML was regressed across trials against power P (normalized envelope) of each frequency band, averaged over time between –4 and 0 s before choice onset:

$$TML = \alpha + \beta P + \epsilon$$

with α corresponding to the intercept and ε to the error term. The significance of the regression estimates β was assessed across recording sites using two-sided, one-sample, Student's t-tests.

In order to determine whether other frequency bands provided additional information relative to the BGA, the following 16 GLMs were compared:

$$TML = \beta_{BGA} \times BGA$$

$$TML = \beta_{BGA} \times BGA + \beta_\gamma \times P\left(\gamma\right)$$

$$TML = \beta_{BGA} \times BGA + \beta_\beta \times P\left(\beta\right)$$

$$TML = \beta_{BGA} \times BGA + \beta_\alpha \times P\left(\alpha\right)$$

$$TML = \beta_{BGA} \times BGA + \beta_\theta \times P\left(\theta\right)$$

$$TML = \beta_{BGA} \times BGA + \beta_\gamma \times P\left(\gamma\right) + \beta_\beta \times P\left(\beta\right)$$

$$TML = \beta_{BGA} \times BGA + \beta_\gamma \times P\left(\gamma\right) + \beta_\alpha \times P\left(\alpha\right)$$

$$TML = \beta_{BGA} \times BGA + \beta_\gamma \times P\left(\gamma\right) + \beta_\theta \times P\left(\theta\right)$$

$$TML = \beta_{BGA} \times BGA + \beta_\beta \times P\left(\beta\right) + \beta_\alpha \times P\left(\alpha\right)$$

$$TML = \beta_{BGA} \times BGA + \beta_\beta \times P\left(\beta\right) + \beta_\theta \times P\left(\theta\right)$$

$$TML = \beta_{BGA} \times BGA + \beta_\alpha \times P\left(\alpha\right) + \beta_\theta \times P\left(\theta\right)$$

$$TML = \beta_{BGA} \times BGA + \beta_\gamma \times P\left(\gamma\right) + \beta_\beta \times P\left(\beta\right) + \beta_\alpha \times P\left(\alpha\right)$$

$$TML = \beta_{BGA} \times BGA + \beta_\gamma \times P\left(\gamma\right) + \beta_\beta \times P\left(\beta\right) + \beta_\theta \times P\left(\theta\right)$$

$$TML = \beta_{BGA} \times BGA + \beta_\gamma \times P\left(\gamma\right) + \beta_\alpha \times P\left(\alpha\right) + \beta_\theta \times P\left(\theta\right)$$

$$TML = \beta_{BGA} \times BGA + \beta_\beta \times P\left(\beta\right) + \beta_\alpha \times P\left(\alpha\right) + \beta_\theta \times P\left(\theta\right)$$

$$TML = \beta_{BGA} \times BGA + \beta_\gamma \times P\left(\gamma\right) + \beta_\beta \times P\left(\beta\right) + \beta_\alpha \times P\left(\alpha\right) + \beta_\theta \times P\left(\theta\right)$$

with β corresponding to the regression estimates, and P each power time series averaged between –4 and 0 s before choice onset in the high-gamma (BGA), gamma (γ), beta (β), alpha (α), and theta (θ) frequency bands.

The model comparison was conducted using the Variational Bayesian Analysis (VBA) toolbox (*Daunizeau et al., 2014*). Log-model evidence obtained in each recording site was taken to a group-level, random-effect, Bayesian model selection (RFX-BMS) procedure (*Rigoux et al., 2014*). RFX-BMS provides an exceedance probability that measures how likely it is that a given model is more frequently implemented, relative to all the others considered in the model space, in the population from which samples were drawn.

## Computational analysis of baseline activity effect on choices

For each recording site of a given ROI, BGA was averaged in each trial over the window corresponding to the best significant cluster obtained with our second GLM (regression of residual error of choice model fit against power). More specifically, activity was averaged from –1.64 to –1.31 s before choice onset for the vmPFC and from –0.95 to –0.67 s for the daIns. The choice model was then run separately with data from trials whose averaged activity was above or below the median baseline activity across trials. As we expected small effects, data were restricted to trials that were not overly determined by choice parameters (i.e. trials for which acceptance probability, as computed by the choice model with all trials, was between 2/7 and 5/7 of the subject's acceptance range). We also ensured that the mean and variance of choice dimensions (gain, loss, and difficulty) were identical between our two trial subsets (high vs. low baseline BGA). All free parameters of the expected utility were free to fluctuate (the constant $k_0$, gain weight $k_g$, loss weight $k_l$, and the weight associated with difficulty σ), while $k_t$ was

set with values obtained from the previously computed model (with all trials). Finally, significance of the difference between posterior parameters obtained with the two trial subsets (high vs. low activity) was tested across all contacts of a given region using two-sided, paired, Student's t-tests.

## Code availability

The custom codes used to (i) extract the different frequency envelopes, and in particular the BGA, from the raw intracranial data, (ii) perform the regression analyses at recording site level, and (iii) compute the second level statistics (across all recording sites of a ROI) are available at: https://gitlab.com/romane-cecchi/publications-code/2022-ieeg-mood-and-risky-choice, (copy archived at swh:1:rev:8b-ccf85407231f5a7d9e117e8e2656a9cb43c083; *Cecchi, 2022*).

## Acknowledgements

This work benefited from the program from University Grenoble Alpes, within the program 'Investissements d'Avenir' (ANR-17-CE37-0018; ANR-18-CE28-0016; ANR-13-TECS-0013). JH and PM were funded GACR (grant number 20–21339 S). The funders had no role in study design, data collection and analysis, decision to publish or preparation of the manuscript.

## Additional information

### Funding

| Funder | Grant reference number | Author |
| --- | --- | --- |
| Université Grenoble Alpes | ANR-17-CE37-0018 | Julien Bastin |
| Université Grenoble Alpes | ANR-18-CE28-0016 | Julien Bastin |
| Université Grenoble Alpes | ANR-13-TECS-0013 | Philippe Kahane Julien Bastin |
| Czech Science Foundation | 20-21339S | Jiri Hammer |

The funders had no role in study design, data collection and interpretation, or the decision to submit the work for publication.

### Author contributions

Romane Cecchi, Data curation, Formal analysis, Investigation, Visualization, Methodology, Writing - original draft, Project administration, Writing - review and editing; Fabien Vinckier, Conceptualization, Methodology, Writing - original draft; Jiri Hammer, Investigation; Petr Marusic, Anca Nica, Sylvain Rheims, Agnès Trebuchon, Emmanuel J Barbeau, Marie Denuelle, Louis Maillard, Lorella Minotti, Philippe Kahane, Resources; Mathias Pessiglione, Conceptualization, Supervision, Validation, Writing - original draft, Writing - review and editing; Julien Bastin, Conceptualization, Supervision, Funding acquisition, Validation, Methodology, Writing - original draft, Project administration, Writing - review and editing

### Author ORCIDs

Romane Cecchi (iD) http://orcid.org/0000-0002-6149-939X
Agnès Trebuchon (iD) http://orcid.org/0000-0002-8632-3454
Emmanuel J Barbeau (iD) http://orcid.org/0000-0003-0836-3538
Julien Bastin (iD) http://orcid.org/0000-0002-0533-7564

### Ethics

Human subjects: All patients gave written, informed consent before their inclusion in the present study, which received approval from the local ethics committees (CPP 09-CHUG-12, study 0907; CPP18-001b / 2017-A03248-45; IRB00003888; CER No. 47-0913).

### Decision letter and Author response

Decision letter https://doi.org/10.7554/eLife.72440.sa1
Author response https://doi.org/10.7554/eLife.72440.sa2

# Additional files

## Supplementary files
- Supplementary file 1. Tables of demographic data and statistical results.
- Transparent reporting form
- Source data 1. Estimates obtained from the regression of BGA against TML in vmPFC and daIns. Rows: time course, columns: recording sites.
- Source data 2. Estimates obtained from the regression of choice residuals against BGA in vmPFC and daIns. Rows: time course, columns: recording sites.

## Data availability
Due to ethical restrictions on data sharing, we are unable to share raw data for this manuscript to preserve participant anonymity. However, anonymized iEEG data in BIDS format can be made available upon request to the corresponding author (JB) and source data files with anonymized regression estimates are available for download. The custom codes used to (i) extract the different frequency envelopes, and in particular the broadband gamma activity (BGA), from the raw intracranial data, (ii) perform the regression analyses at recording site level, and (iii) compute the second level statistics (across all recording sites of a ROI) are available at: https://gitlab.com/romane-cecchi/publications-code/2022-ieeg-mood-and-risky-choice (copy archived at swh:1:rev:8bccf85407231f5a7d9e117e8e2656a9cb43c083).

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
