## [Editor Report]

This study used intracranial EEG to explore links between broad-band γ oscillations and mood, and how they impact risky decision-making. The topic is interesting for cognitive neuroscientists and researchers interested in computational psychiatry.

---

## [Decision Letter]

**Decision letter after peer review:**

Thank you for submitting your article "Intracerebral mechanisms explaining the impact of incidental feedback on mood state and risky choice" for consideration by *eLife*. Your article has been reviewed by 3 peer reviewers, and the evaluation has been overseen by a Reviewing Editor and Michael Frank as the Senior Editor. The reviewers have opted to remain anonymous.

Reviewers agreed that your study is potentially interesting and that with appropriate analysis and revisions, the results are likely to be of interest to a broad audience. However, there were fundamental concerns about some aspects of your study, including the statistical approach. It will be important that you fully address these essential issues, as the reviewers' confidence in your findings will in large part rest on your responses to these concerns.

Essential revisions:

1. Reviewers were concerned about a potential confound in the task design that may explain results. In 25% of the trials, subjects were asked to provide a mood rating, whereas in the remaining 75% of trials, the screen was just blank and the subjects rested. Trials without mood ratings would be expected to trigger increased BGA activity in default mode areas such as the vmPFC, and opposite patterns in salience, visual and motor areas. It would be important to compare and plot "raw" BGA between trials with and without mood ratings, and discuss whether and if so how default mode activity induced by the rest period could have affected the results.

2. The analyses focus on the vmPFC and insula, which is understandable given previous work that has implicated these regions in mood. However, it appears that several areas show similar correlations, and it would be important to also analyze and report results from these other areas to show whether or not effects are specific for the vmPFC and insula.

3. Reviewers were concerned about the statistical analysis. Specifically, the authors grouped all data across all subjects and performed statistics across electrodes. Reviewers considered this approach invalid as it inflates the number of independent observations (electrodes from the same subject are not independent). It would be better to compute statistics across subjects per region. Please revise the statistical approach throughout the manuscript by replacing any and all across-electrode analyses with across-subjects analyses, with degrees of freedom reflecting the number of subjects with contacts in each particular region (one observation per subject, averaged across electrodes). In addition, it would be good to report how many subjects show statistically significant regressions between BGA and mood at any electrode.

4. There was considerable discussion among reviewers about details of the design and analyses. For instance, it was not immediately clear for which trials the regression analyses for mood ratings and TML were computed. It would be important to revise and clarify the methods throughout the manuscript.

---

## [Author Response]

Essential revisions:1. Reviewers were concerned about a potential confound in the task design that may explain results. In 25% of the trials, subjects were asked to provide a mood rating, whereas in the remaining 75% of trials, the screen was just blank and the subjects rested. Trials without mood ratings would be expected to trigger increased BGA activity in default mode areas such as the vmPFC, and opposite patterns in salience, visual and motor areas. It would be important to compare and plot "raw" BGA between trials with and without mood ratings, and discuss whether and if so how default mode activity induced by the rest period could have affected the results.

We understand the suggestion that during rest periods (no mood rating), activity would increase in “default mode” regions such as the vmPFC, whereas during mood rating periods, activity would increase in “salience” regions, such as the daIns. However, we do not see how this possibility could confound the results (i.e., the association of BGA with mood level), simply because the distribution of rating vs. rest trials was orthogonal to the distribution of positive vs. negative feedback trials that created mood fluctuations. There could be a potential confound, should the presence of a mood rating by itself affect TML, but this was not the case, because by construction, the presence of a rating was not integrated in the computation of TML.

A way to definitively rule out this possibility is to test the correlation between mood and BGA across rating (no rest) trials only. Even if the presence of a rating by itself affects mood or BGA, it would do so in every trial and therefore cannot bias the correlation across trials. This analysis was already provided in the initial manuscript (see Figure 2b). It shows that the association between mood and BGA in both vmPFC and daIns holds even when restricted to rating trials and therefore cannot be confounded by the absence of rating (in rest trials) inducing default mode activity in the vmPFC.

To better explain how ratings and feedbacks were distributed across trials, we have added a supplementary figure that shows a representative example (Figure 1—figure supplement 1). This plot shows that ratings were collected independently of whether subjects were in high- or low-mood episodes.

We suspect the comment relates to the recent publication by the Parvizi group (Kucyi et al., 2020) reporting antagonist BGA in the default mode network (which includes vmPFC) and the saliency network (which includes daIns). Even if this anticorrelation between vmPFC and daIns was indeed triggered in our data by the alternance of rating and no-rating trials, it could not affect our results, first for the reasons explained above (this alternance is orthogonal to mood fluctuations) and second because vmPFC and daIns explained different parts of the variance in mood level (see Figure 2C of the initial manuscript). In other words, the mood-relevant information carried by vmPFC and daIns was not the same, and therefore was not a simple reflection of the same factor (presence vs. absence of mood rating). This was already explained in the initial manuscript, but we now better emphasize this point, in the revised discussion, with citation of the paper by Kucyi and colleagues (lines 440-447):

“Conversely, including low-frequency activity in the model did not help predicting mood level. The information carried in low-frequency activity was therefore at best redundant with that of high-frequency activity, whereas the two regions carried at least partly independent information. When analyzing separately bad, neutral, and good mood levels, we found that higher BGA signaled the two extreme tertiles: good mood in the vmPFC and bad mood in the daIns. This result suggests that the two regions were specifically concerned with good and bad mood, explaining why they were not just anti-correlated, as previously suggested (Kucyi et al., 2020).”

2. The analyses focus on the vmPFC and insula, which is understandable given previous work that has implicated these regions in mood. However, it appears that several areas show similar correlations, and it would be important to also analyze and report results from these other areas to show whether or not effects are specific for the vmPFC and insula.

Following this suggestion, we now provide a more systematic exploration of the dataset, with a detailed analysis of other brain regions for all three main parametric analyses testing the association between BGA and (i) mood ratings, (ii) TML and (iii) choice residuals. Interestingly, only the vmPFC was positively related to both choice and mood (whether proxied by rating or TML), and only the daIns was negatively associated with both choice and mood (whether rating or TML). Thus, our two ROI were the only ones meeting the criteria for mediating the impact of mood on choice, which we show in subsequent analyses.

We now systematically provide detailed results for all significant associations (both positive and negative) with each parametric regressor (mood ratings, TML, choice residuals) in Tables S2-S3 in Supplementary file 1.

We edited the “mood” Results section (lines 236-246) as follows:

“The vmPFC (n = 91 sites from 20 subjects) was the only ROI for which we found a positive correlation (Figure 2b; Source data 1; Table S2 in Supplementary file 1) between BGA and both mood rating (best cluster: -1.37 to -1.04 s, sum(t_(90)_) = 122.3, p_corr_ = 0.010) and TML (best cluster: -0.57 to -0.13 s, sum(t_(90)_) = 132.4, p_corr_ = 8.10^-3^). Conversely, we found a negative correlation in a larger brain network encompassing the daIns (n = 86 sites from 28 subjects, Figure 2b; Source data 1; Table S2 in Supplementary file 1), in which BGA was negatively associated with both mood rating (best cluster: -3.36 to -2.51 s, sum(t_(85)_) = -325.8, p_corr_ < 1.7.10^-5^) and TML (best cluster: -3.13 to -2.72 s, sum(t_(85)_) = -136.4, p_corr_ = 9.10^-3^). The additional brain regions in which BGA was negatively associated with mood level (indexed with either mood rating or TML) involved the superior portion of the dorsolateral prefrontal cortex, the lateral and rostro-mesial visual cortex, the ventral motor cortex and the dorsomedial premotor cortex (Table S2 in Supplementary file 1).”

We also edited the “choice” Results section (lines 320-324) as follows:

“The brain regions showing a significant positive association between choice and BGA included the mid-cingulate, dorsal, rostral and ventral medial prefrontal cortex, the ventral anterior insula, the rostral inferior temporal cortex and parts of the visual and motor cortex. Negative association between choice and BGA was observed in regions such as the rostro-ventral premotor cortex, the posterior and dorsal anterior insula, the ventral inferior and the superior parietal cortex (Table S3 in Supplementary file 1).”

In the supplementary materials, we added the following two tables:

3. Reviewers were concerned about the statistical analysis. Specifically, the authors grouped all data across all subjects and performed statistics across electrodes. Reviewers considered this approach invalid as it inflates the number of independent observations (electrodes from the same subject are not independent). It would be better to compute statistics across subjects per region. Please revise the statistical approach throughout the manuscript by replacing any and all across-electrode analyses with across-subjects analyses, with degrees of freedom reflecting the number of subjects with contacts in each particular region (one observation per subject, averaged across electrodes). In addition, it would be good to report how many subjects show statistically significant regressions between BGA and mood at any electrode.

We respectfully disagree with the idea that performing statistics across recording sites is invalid. What the reviewers suggest is the group-level random-effect analysis that is typically used in fMRI studies but is not the gold standard in electrophysiology studies. Should fixed-effect analyses, pooling recording sites across subjects, be invalid, the quasi-totality of single-cell recording studies in monkeys, as well as iEEG studies in humans, should be withdrawn from the scientific literature. There are two main reasons, in our opinion, why the electrophysiology community has converged on a fixed-effect approach. The first is the low number of participants (typically 2 or 3 in monkey studies, and less than 10 in most iEEG studies until recently), which would make group-level analyses severely under-powered. The second is the inhomogeneous sampling (number and location of recording sites in a given region varies a lot across patients), such that giving the same weight to every participant would severely bias statistical estimates. Thus, pooling recording sites across subjects (or monkeys) ensure that the region of interest is reasonably sampled in terms of spatial coverage and statistical power.

That being said, we agree that pooling recording sites across subjects (or monkeys) is making the implicit assumption that differences between brains are negligible, which is not satisfactory, even if standard in the field. To address this issue, we reanalyzed the entire dataset using mixed-effect models, which distinguish variance across recording sites and differences between individuals.

Specifically, we tested in each ROI the association between BGA (averaged across the best temporal cluster identified with the fixed-effect approach) and (1) mood ratings, (2) TML and (3) choices with the following linear mixed models:

BGA ~ 1 + mood + (1 + mood | SUBJECTS) + (1 + mood | RECORDING SITES:SUBJECTS)BGA ~ 1 + TML + (1 + TML | SUBJECTS) + (1 + TML | RECORDING SITES:SUBJECTS)Choice ~ 1 + BGA + (1 + BGA | SUBJECTS) + (1 + BGA | RECORDING SITES:SUBJECTS)

We found that all effects remained significant in our two main ROIs (vmPFC and daIns), with the same sign as in the initial fixed-effect analyses, while other regions showed associations passing the significance threshold for the other regressors (see Tables S2 and S3 in Supplementary file 1).

These analyses have been incorporated into the Results section (lines 248-251), as follows:

“To avoid making the implicit assumption, related to fixed-effect analyses across recording sites, that differences between brains are negligible, we checked that these effects remained significant using generalized linear mixed-effects models, which separately account for inter-site and inter-individual variance (Tables S2-3 in Supplementary file 1).”

The methods section (lines 821-832) has also been modified, accordingly:

“Group-level analyses

To avoid making the implicit assumption, related to fixed-effect analyses across recording sites, that differences between brains are negligible, we checked that these effects remained significant using mixed-effects linear models, which separately account for inter-site and inter-individual variance (Tables S2-3 in Supplementary file 1).

Specifically, we tested in each ROI the association between mean BGA (averaged over the best temporal cluster identified in fixed-effects analyses) and (1) mood rating, (2) TML and (3) choice with the following linear mixed-effects models:

BGA ~ 1 + mood + (1 + mood | SUBJECTS) + (1 + mood | RECORDING SITES:SUBJECTS)BGA ~ 1 + TML + (1 + TML | SUBJECTS) + (1 + TML | RECORDING SITES:SUBJECTS)Choice ~ 1 + BGA + (1 + BGA | SUBJECTS) + (1 + BGA | RECORDING SITES:SUBJECTS)

We then performed F-tests on the fixed effects estimates, with significance set at p < 0.05, two-tailed.”

Finally, to satisfy the reviewers, we also tested these associations with group-level random-effects analyses, where data points are subject-wise BGA averaged across recording sites (within the temporal cluster identified with the fixed-effects analyses). All associations remained significant or borderline (see Table S4 in Supplementary file 1), even with this under-powered approach. For all the reasons detailed above, we prefer to incorporate these results in a supplementary table and keep the fixed-effects analyses for the main results, as they will be better accepted by the iEEG community.

These analyses have been incorporated into the Results section (lines 356-359), as follows:

“We also verified that the main findings of this study remained significant (or borderline) when using group-level random-effects analyses (Table S4 in Supplementary file 1, see methods), even if this approach is underpowered and unadjusted for sampling bias (some subjects having very few recording sites in the relevant ROI).”

The methods section has also been edited, as follows (lines 833-837):

“To test the association between BGA and mood, TML or choice at the group level (using random-effects analyses), we performed the same linear regression as described in the electrophysiological analyses section on BGA averaged over the best time cluster (identified by the fixed-effects approach) and across all recording sites of a given subjects located in the relevant ROI. We then conducted a two-sided, one-sample Student's t-test on the resulting regression estimates (Table S4 in Supplementary file 1).”

4. There was considerable discussion among reviewers about details of the design and analyses. For instance, it was not immediately clear for which trials the regression analyses for mood ratings and TML were computed. It would be important to revise and clarify the methods throughout the manuscript.

We apologize for this lack of clarity. We have checked that this issue is now fixed in the revised manuscript. Note that regarding the specific point raised, the regression against mood rating is necessarily done over trials with mood rating, so there cannot be any ambiguity there. Regarding the regression against TML, it was done both over the trials with mood rating only and over all trials (with and without mood rating), as indicated above the corresponding plots.